# Ax-Prover: Agentic Lean Proving with LLMs and MCP Integration

## Abstract

We present Ax-Prover, a domain-agnostic multi-agent system for automated theorem proving in Lean. Formal proof generation requires both creative reasoning and strict syntactic rigor. Ax-Prover meets this challenge by combining large language models (LLMs), which provide knowledge and reasoning, with MCP Lean tools, which ensure correctness. To evaluate performance, we benchmark our approach on the large-scale NuminaMath-LEAN dataset and introduce two new Lean benchmarks in Abstract Algebra and Quantum Theory. Across all domains, Ax-Prover outperforms state-of-the-art provers, with particularly large gains in the new benchmarks – indicating that while Ax-Prover adapts readily to novel areas, existing systems remain narrowly specialized and struggle to generalize.

## 1 Introduction

Large Language Models (LLMs) have become the standard approach to address complex tasks in both academic fields and industry. A relevant application of such models is in the field of mathematics, where they have been used to solve complex problems achieving outstanding performance (Chervonyi et al., 2025). More recently, considerable effort has been put into training LLMs to perform formal theorem proving using Lean (de Moura et al., 2015), an open-source programming language and proof assistant that, together with its community-driven `mathlib` library (Lean Prover Community, 2025), provides a rigorous setting where AI systems must engage with symbolic reasoning, structured formalization, and evolving mathematical knowledge. This makes Lean an attractive testbed for probing the reasoning capabilities of LLMs.

Recent LLM provers such as the DeepSeek-Prover series (Xin et al., 2024a; DeepSeek-AI, 2024; Ren et al., 2025), Kimina-Prover-72B (Wang et al., 2025), Goedel-Prover Lin et al. (2025), and Seed-Prover Chen et al. (2025) have shown that distillations of frontier reasoning models or RL based training, when adapted for theorem proving in Lean, can reach state-of-the-art performance on benchmarks like Mini-F2F (Zheng et al., 2021) and Putnambench Tsoukalas et al. (2024). Despite these results, they face several key limitations. First, their ability to generalize beyond the Lean mathematics distributions on which they were trained remains unclear, which limits their broader applicability. Relatedly, they depend on fixed versions of the fast-evolving `mathlib`, making them brittle to new definitions unless continuously re-trained, which adds significant cost. Second, it is hard to run them, as they require high-spec machines and expertise to be successfully deployed and used. Third, while distillation improves their ability to produce Lean proofs, it narrows their capabilities: compared to their parent reasoning models, they lose tool use and interactive abilities, limiting effective human–AI collaboration. Together, these issues suggest that scaling increasingly large, specialized provers may yield diminishing returns in both flexibility and usability.

In contrast, general-purpose frontier LLMs like Claude (Anthropic, 2025b) and GPT (OpenAI, 2025) encode substantial prior knowledge across a variety of domains (e.g., mathematics, physics, and computer science), while also exhibiting strong natural language understanding, problem-solving skills, and interaction capabilities. Yet, they are not explicitly trained to formalize statements or construct proofs in Lean and thus cannot natively engage with the Lean environment. This creates a sharp division: specialized provers are tightly integrated with Lean but narrow and hard to use, whereas general-purpose models are broad and easily accessible but lack the ability to access the formal reasoning infrastructure required for theorem proving.

To address this gap, we introduce Ax-Prover,[1] a new agentic workflow for Lean theorem proving that equips general-purpose LLMs with direct access to the Lean proof environment through external tools (Dressler, 2025a). Ax-Prover enables LLMs to reason about unproven theorems, propose proof sketches, and generate step-by-step Lean code, while using the lean-lsp-mcp (Dressler, 2025a;b) to inspect goals, search for relevant results, locate errors, and verify code

---

[1]"Ax" stands for "axiomatic", indicating the base principles in mathematics and physics, the domains explored in this work.

– capabilities essential for formal theorem proving. Ax-Prover overcomes the main limitations of current state-of-the-art provers. First, it avoids domain overspecialization and obsolescence tied to fixed `Mathlib` versions. Second, by leveraging existing frontier models, it sidesteps the need to host and deploy specialized systems. Third, it preserves tool-use and conversational abilities, enabling interactive collaboration with both experts (for targeted feedback) and non-experts (for guidance and advice).

We evaluated Ax-Prover on three datasets. The first is `NuminaMath-LEAN`, an established benchmark of mathematics competition problems. To broaden the evaluation, we also introduce two new datasets. **AbstractAlgebra** focuses on algebraic structures such as groups, rings, and fields, testing the prover's ability to reason in a more abstract, research-oriented setting rather than the competition-driven style of `NuminaMath`. **QuantumTheorems** explores the domain of quantum physics, assessing whether the prover can extend beyond pure mathematics and transfer its reasoning to scientific applications. Our experiments show that Ax-Prover outperforms general-purpose LLMs not equipped with Lean tools as well as state-of-the-art specialized provers across the board, gaining flexibility and robustness without sacrificing accuracy.

Our contributions are twofold: (1) We design **Ax-Prover**, a lightweight agentic workflow that connects general-purpose LLMs to Lean via lean-lsp-mcp, and demonstrate it outperforms both general-purpose LLMs and specialized provers on competition-level mathematics, abstract algebra, and quantum physics; (2) We contribute new formalized **Lean datasets** covering physics and abstract algebra, complementing existing benchmarks.

## 2 RELATED WORK

Automated theorem proving in Lean has roots in classical approaches such as decision procedures (de Moura & Bjørner, 2008; Barbosa et al., 2022) and heuristic-guided proof search (Kovács & Voronkov, 2013; Schulz et al., 2019), but they face scalability challenges and rarely generate proofs in a form usable by mathematicians. More recent work integrates machine learning: from heuristic tuning (Urban et al., 2011) to premise selection and tactic prediction (Irving et al., 2016; Huang et al., 2019), culminating in transformer-based language models capable of generating Lean proofs (Polu & Sutskever, 2020; Lample & Charton, 2022; Polu et al., 2023; Xin et al., 2024b). Recent large-scale systems extend this trend by distilling and fine-tuning massive base models (e.g., Kimina from Qwen2.5-72B (Wang et al., 2025); DeepSeek-Prover-V2 from DeepSeek-V3 (Ren et al., 2025)). These pipelines achieve impressive performance but demand enormous GPU resources and engineering efforts, producing specialized provers that don't generalize across domains. Also, Mathlib's rapid growth – now containing over 220,000 theorems and adding thousands more each month (Lean Prover Community, 2025) – highlights the need for tools that are both efficient and adaptable to evolving mathematical libraries. Moreover, these models usually cannot engage in collaboration with human experts: given an input theorem, they move on straight to its formalization and proof. This is the main approach among current provers, implemented also in Lin et al. (2025), Baba et al. (2025), and Ospanov et al. (2025).

A parallel line of work has explored classical machine learning for supporting experts in Lean proving, for example in premise selection and tactic prediction (Gauthier et al., 2021; Blaauwbroek et al., 2020), and more recently through LLMs that connect to Lean via external interfaces (Ayers et al., 2023; Azerbayev et al., 2023; Song et al., 2024). These approaches illustrate the promise of AI-assisted proving, but they remain resource-intensive and difficult to adapt across domains. Recent efforts, such as Kumarappan et al. (2024), attempt to remedy this by emphasizing greater adaptability within Lean. At the same time, there is growing interest in human–AI collaboration: conversational assistants (Collins et al., 2023) and "copilot"-style integrations (Chen et al., 2021) suggest how formal tools can augment, rather than replace, human reasoning. Our work builds on this trajectory by closing the gap between heavyweight, specialized provers and lightweight, researcher-friendly systems that can be more readily adapted to the evolving Lean ecosystem.

## 3 SYSTEM ARCHITECTURE

We implement **Ax-Prover** as a modular multi-agent architecture with three specialized agents: the **Orchestrator**, the **Prover**, and the **Verifier**. Following recent agentic designs for complex tasks such as scientific discovery (Gottweis et al., 2025; Yamada et al., 2025), we avoid a monolithic setup by assigning each sub-agent[2] a distinct role. This separation enables specialization and modularity: agents can be independently optimized, replaced, or extended allowing researchers to adapt Ax-Prover to new domains, integrate additional tools, or tailor workflows to their own expertise without destabilizing the system.

---

[2]We use "sub-agent" and "specialized agent" interchangeably.

Our workflow is straightforward. The Orchestrator receives an unproven Lean statement and forwards it to the Prover, which iteratively generates a proof. The Verifier then checks the proof and reports back. If valid, the Orchestrator closes the task; otherwise, it provides feedback to the Prover, which attempts a new proof. Through this closed-loop process, the system incrementally converts unproven theorems into formally verified Lean proofs. We next describe each agent in detail and the tools underlying our workflow.

## 3.1 Specialized Agents

### 3.1.1 Orchestrator

The Orchestrator's role is analogous to a scheduler in distributed systems: it does not perform computation itself but ensures that computation flows smoothly across agents. It holds three main responsibilities. First, it handles **task assignment**, as it receives user input and instructs the Prover accordingly. Next, it manages **feedback routing** by taking diagnostic outputs from the Verifier and giving structured feedback to the Prover. This separation ensures that proof synthesis and evaluation remain distinct while still enabling iterative refinement. Finally, it decides when to **stop the refinement loop**. Termination occurs either when the Verifier certifies the proof as complete and error-free, or when repeated failures exceed a configurable threshold, deeming the theorem irrecoverable under the given resource budget.

### 3.1.2 Prover

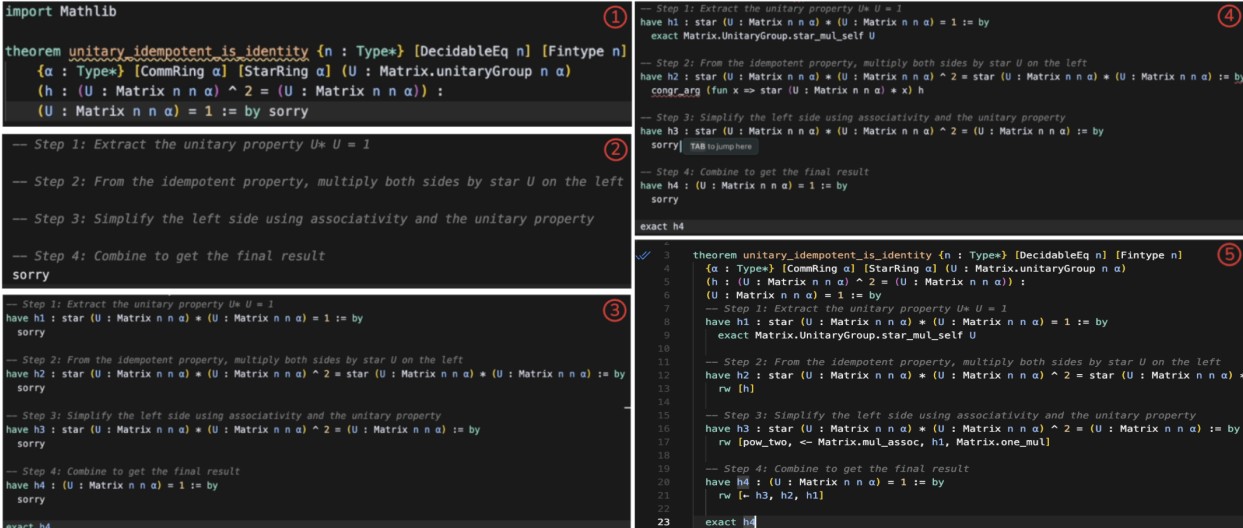

Figure 1: Prover steps.

The Prover is the constructive core of the system. Its task is to transform unproven Lean theorems into completed proofs. Theorem proving requires both creativity – finding the right lemma or using the right tactic – and discipline – ensuring that the structure and Lean code are syntactically correct. To achieve this, the Prover balances LLM-based heuristic exploration with rigorous Lean formalization aided by the Lean tools made available by the lean-lsp-mcp (see Section 3.2).

We instruct the Prover to carry out its task following this general approach. Note that at each stage, the Prover writes the updated version of the theorem to a `.lean` file. This is for two reasons: first, it complies with Lean tools requirements, which require filepaths to function. Second, writing to a file at each step allows the user to inspect the proving process in real time (see Figure 1):

1. **Theorem identification:** The Prover scans Lean files for unfinished proofs marked with `sorry` - for example, the `Unitary Idempotent is Identity` theorem in box 1 in Figure 1. This step guarantees that the prover attempts to prove all valid theorem statements.

2. **Sketch construction:** Next the Prover generates a coarse-grained natural language outline of the proof's logical flow (box 2), breaking down a complex proof into more manageable steps, and briefly describing the

key ideas for each step (e.g., algebraic manipulation). This mirrors how human mathematicians would break down a complex problem, starting with a high-level sketch before filling in the details.

3. **Stepwise formalization:** Then the Prover formalizes each of the steps in Lean (box 3). Each formalized step starts with and `have` and ends with a `sorry`.

4. **Sequential Solving:** The Prover goes through each step sequentially, proposing Lean tactics to substitute the `sorry` (box 4).

5. **Verification check:** After completing each step, the Prover uses `lean_diagnostic_messages` to assess if the generated step is correct. If a severity 1 error is detected, the Prover halts progress, backtracks, and attempts an alternative strategy. If the check is correct and there are no `sorry`s left, the Prover ends its task (box 5).

This approach allows the Prover to function like an automated yet cautious mathematician: it incrementally explores and implements ideas, verifies their correctness in Lean, and advances only once each step has been validated.

### 3.1.3 VERIFIER

The Verifier serves as the final gatekeeper of correctness in our workflow. It neither generates nor modifies proofs: it only assesses the correctness of the proof generated by the Prover. Similarly to the Prover, the Verifier has access to filesystem tools – required to access the file produced by the Prover – and a single Lean tool, `lean_diagnostic_messages`, to assess the correctness of the proof.

Concretely, the Verifier operates in two steps: First, it **compiles** the Lean file produced by the Prover using `lean_diagnostic_messages`. Second, it parses the diagnostic message, generate a reports of the errors, and **emits a verdict**: a proof is considered verified if and only if no level-1 error exists and there are no sorrys or admit statements present (see Section 3.2).

At first glance the Verifier may seem redundant, since it uses the same `lean_diagnostic_messages` tool as the Prover. However, it is needed for two reasons: (i) the Prover may run out of steps and return an incomplete or incorrect proof, and (ii) it sometimes terminates early despite remaining errors. An independent Verifier thus ensures robustness, mirroring software pipelines where aggressive testing is always checked by a conservative compiler.

### 3.2 MCP TOOLS

As mentioned above, tools are essential for the Prover to complete a proof. We provide tool access via the MCP protocol, a standard interface that lets LLM agents invoke external services in a uniform, controlled way. The Prover uses two categories of tools: **Filesystem tools** and **Lean tools**. Filesystem tools handle file operations such as `read_file`, `write_file`, and `list_directory` (see Appendix A.1). Lean tools allow the Prover to perform a variety of actions crucial for theorem proving. We access these tools through the lean-lsp-mcp project Dressler (2025a), which provides a standardized interface to the Lean environment and ensures that the Prover always operates on the latest version of `Mathlib`, maintaining compatibility for imports, theorem references, and proof construction. The tools themselves fall into four main groups, summarized in Table 1.

| Category | Tools |
|---|---|
| Project and File Management | `build`: Compile and build the Lean project 
 `file_contents`: Get contents of a Lean file 
 `declaration_file`: Find which file contains a declaration |
| Diagnostics and Feedback | `lean_diagnostic_messages`: Compile code and return diagnostic messages 
 `goal`: Get the current proof goal at a position 
 `term_goal`: Get goal information for a term 
 `hover_info`: Get hover information for symbols |
| Code Assistance | `completions`: Get completion suggestions 
 `multi_attempt`: Try multiple proof attempts 
 `run_code`: Execute Lean code |
| Search and Reasoning | `leansearch`: Search for theorems and lemmas 
 `loogle`: Search for lemmas by type signature 
 `state_search`: Search proof states 
 `hammer_premise`: Use automated theorem proving |

Table 1: Lean tools available on `lean-lsp-mcp`, organized by functionality.

Note that `lean_diagnostic_messages` returns a diagnostic with the error log and a scalar: 0 if no error is found; 1 for incorrect/incomplete proofs; and 2 for a valid but incomplete proof, e.g. with `sorry`, warning, or linter error.

## 4 DATASETS

While the application of LLMs to mathematical verification in Lean is evolving rapidly, the availability of comprehensive datasets remains limited. At present, only a few open-source datasets are available, with some of the most notable being **MiniF2F** (Zheng et al., 2021), **Putnambench** Tsoukalas et al. (2024), and **NuminaMath-LEAN** (Numina-Team, 2025). These benchmarks include hard, high-level math problems from competitions such as the International Mathematical Olympiad (IMO) or the Putnam exam. Other datasets exist, but have clear limitations: For example, the Deepseek-Prover-V1 Train(DeepSeek-AI, 2024) includes 27k LLM-generated statements and proofs, but most of them are very simple, with 2–3 line proofs. Also Lean Workbook(Ying et al., 2024) (57k) gathers LLM-generated formalizations of mathematical problems. While it reports a $93.5\%$ statement-level accuracy after filtering, subsequent analyses note that a nontrivial fraction of examples still suffer from semantic errors and hallucinations (Lu et al., 2025; Wu et al., 2025), which limits its reliability.

All in all, current valuable datasets (MiniF2F, Putnambench, and NuminaMath-LEAN), not only cover a single domain - mathematics - but they also focus on a very specific type of math problem, i.e, competition-level problems. To enrich the current ecosystem and expand the coverage of Lean datasets, we create and release two new datasets.

**Abstract-Algebra** (**AA**) is a Lean 4 dataset of problems drawn from standard abstract algebra textbooks. Unlike MiniF2F (Zheng et al., 2021), Putnambench Tsoukalas et al. (2024), and NuminaMath-LEAN (Numina-Team, 2025), which focus on undergraduate level competition-style puzzles, AA targets graduate or research-level mathematics, emphasizing deeper abstract concepts over lengthy step-by-step manipulations.

**Quantum-Theorems** (**QT**) covers core topics in foundational quantum mechanics, spanning problems from density matrices to scaling laws for quantum repeater networks. By bridging theoretical physics with formal verification methods, QT offers an unprecedented opportunity to test prover agents outside the field of Math, providing a rigorous testbed for evaluating automated Lean theorem proving systems on quantum mechanical theorems.

In the section below, we provide more information about each dataset that we use for our experiments.

### 4.1 ABSTRACT-ALGEBRA

**Abstract-Algebra** (**AA**) is a curated dataset of 100 Lean problems inspired by exercises in Dummit & Foote's abstract algebra textbook (Dummit & Foote, 2004). The dataset consists of two subsets: 50 easy problems from Chapter 1.1 and 50 intermediate problems from Chapters 1.2–2.5. An in-depth description of the pipeline used to generate AA and examples from AA are provided in the Appendix B.1. While MiniF2F (Zheng et al., 2021), Putnambench Tsoukalas et al. (2024), and NuminaMath-LEAN (Numina-Team, 2025) focus on high school to undergraduate level competition mathematics, the AA dataset is aimed toward research-level mathematics. The key distinction is that competition-level math typically involves elementary content framed as puzzles that require many reasoning steps. For example, a competition problem may ask to determine all positive integers $a, b$ such that

$$\frac{a^2 + b^2}{ab + 1} \in \mathbb{Z},$$

which is conceptually elementary but requires a sequence of clever number-theoretic transformations. In contrast, research-level mathematics involves deeper concepts with fewer reasoning steps per exercise; for instance, an AA problem may ask: *Prove every subgroup of a cyclic group is cyclic.* This example highlights the focus on abstract structures rather than intricate manipulations, echoing the distinction between competition-level and research-level mathematics.

The creation of the AA dataset is motivated by two main reasons. First, abstract algebra is foundational to much of mathematics, providing essential tools for research in number theory, geometry, topology, and beyond—indeed, 22 of the 32 primary mathematics categories on arXiv build upon it (arx, 2025) It also underpins advances outside math, in fields such as cryptography, physics, and chemistry, making it a natural setting for formal reasoning. Second, there is a practical gap between AI-focused formalization efforts, which largely targets elementary mathematics, and the advanced topics studied by research mathematicians. By formalizing problems from standard textbooks, AA bridges this gap and offers a research-oriented dataset that supports deeper mathematical reasoning.

## 4.2 QUANTUM-THEOREMS

**Quantum-Theorems** (**QT**) includes 134 problems spanning core areas of quantum theory. These problems introduce unique challenges, as they require integrating finite-dimensional linear algebra, complex analysis, and matrix theory with quantum principles such as unitarity, Hermiticity, and measurement postulates. This domain-specific knowledge is largely absent from existing prover datasets, making QT a valuable benchmark for testing and advancing formal reasoning in physics.

QT was generated through an iterative human-in-the-loop process, combining automated proof synthesis with expert curation. An automated coding agent first generated proof attempts, producing both complete proofs and partial derivations. A quantum physics expert then reviewed each proof identifying gaps, correcting errors, and standardizing operator definitions. The final dataset replaces these proofs with `sorry` statements.

We generated problems at two levels of difficulty: Basic problems are short (1–10 lines) and often solvable with standard automation (`simp`, `linarith`), e.g., proving that the diagonal entries of a Hermitian matrix are real. Intermediate level problems require 10–50 lines, systematic case analysis, and orchestration of rewrite rules, such as proving simultaneous diagonalization of commuting observables. An in-depth description of the process used to create QT and examples from QT are provided in Appendix B.2.

QT represents a first step toward computer-verified quantum mechanics, addressing the challenge of ensuring correctness in quantum information protocols and algorithms. The dataset has practical importance beyond research: as quantum technologies grow more complex, errors in proofs or hidden assumptions can have serious consequences. For instance, a recent bug in a proof claiming to break lattice-based cryptography—only identified weeks later by experts—illustrates the risks of unchecked reasoning in high-stakes domains (Rousseau, 2024; Chen, 2024). QT provides a resource to begin developing tools which can help in detect these kind of mistakes earlier.

## 4.3 NUMINAMATH-LEAN

NuminaMath-LEAN (Numina-Team, 2025) is a large-scale collection of approximately 104,000 competition-level mathematics problems formalized in Lean 4. The dataset is created by the same research group that developed the Kimina-Prover. They derived NuminaMath-LEAN from NuminaMath 1.5 (LI et al., 2024), with problems drawn from prestigious contests such as the International Mathematical Olympiad (IMO) and the United States of America Mathematical Olympiad (USAMO).

Each problem includes a formal statement in Lean 4, written either by a human annotator (19.3% of the problems) or by an autoformalizer model (80.7%) (Numina-Team, 2025). Out of the total problems, 25% were correctly proved by Kimina-Prover during its RL training phase (`Solved-K`), 11% were proved by humans (`Solved-H`), while the remaining 64% do not have any proof (`Unsolved`) (Wang et al., 2025; LI et al., 2024; Numina-Team, 2025). We analyzed problems across the three groups and observed a clear difficulty gradient: `Solved-K` < `Solved-H` < `Unsolved`. This ordering aligns with the fact that Unsolved problems could not be handled by Kimina-Prover, providing an implicit measure of hardness. For problems with available proofs, this qualitative assessment is further supported quantitatively: Solved-K proofs are shorter on average than Solved-H (98 vs. 155 lines), indicating greater proof complexity.

## 5 EXPERIMENTS

We evaluated Ax-Prover against frontier LLMs and specialized provers using the datasets from Section 4. For AA and QT, we used the full datasets. From NuminaMath-LEAN, we sampled 300 problems – 100 each from `Solved-K`, `Solved-H`, and `Unsolved` – to balance difficulty levels while keeping experiments budget-friendly. We chose to test NuminaMath-LEAN over MiniF2F and Putnambench because they contains similar problems types, and including these benchmarks would have increased costs unnecessarily. Furthermore, Ren et al. (2025) and Wang et al. (2025) both report results on MiniF2F with pass@32 and pass@8192, and on Putnambench with pass@192 for Kimina put and pass@1024 for DeepSeek put which would be too expensive for us to run or compare against using Ax-Prover. The following subsections outline our setup and present the results.

## 5.1 EXPERIMENTAL SETUP

For our experiments, each Ax-Prover agent was powered by Claude Sonnet 4 (Anthropic, 2025a). Since the agentic flow can run indefinitely, we capped Prover API calls at 100,[3] limited Orchestrator–Prover–Verifier loops to 2, and set a 25-minute timeout per theorem. We compare our approach against three strong baselines:

- **Claude Sonnet 4** (**Sonnet**), without access to any tools. This baseline allows us to assess how the same LLM used for our agentic flow performs if used outside the agentic flow, and without access to MCP tools.

- **DeepSeek-Prover-V2-671B** (**DS-Prover**), a specialized Lean prover.

- **Kimina-Prover** (**Kimina**), another specialized Lean prover.

We applied the 25-minutes timeout to all baselines, and run them with pass@1.[4]

## 5.2 RESULTS

| Dataset | Subset | Ax-Prover | Sonnet | DS-Prover | Kimina |
|---------|--------|-----------|--------|-----------|--------|
| NuminaMath | solved-K | 81% | 7% | 48% | 100% |
| | solved-H | 47% | 8% | 14% | 0% |
| | unsolved | 26% | 1% | 18% | 0% |
| | total | 51% | 5% | 28% | 31% |
| AbstractAlgebra | easy | 72% | 10% | 26% | 12% |
| | intermediate | 56% | 6% | 22% | 14% |
| | total | 64% | 8% | 24% | 13% |
| QuantumTheorems | easy | 100% | 54% | 88% | 72% |
| | intermediate | 92% | 18% | 48% | 34% |
| | total | 96% | 40% | 61% | 57% |

Table 2: AX Agent compared to Claude Sonnet 4 and DeepSeek-Prover-V2-671B across algebra and physics domains. Note that results on NuminaMath for Kimina are reported from Numina-Team (2025), and where obtained during its RL training phase with, on average, pass@68.

We compute results by running an external Lean compiler on the files generated by the model – if the file compiles with no sorrys we mark it as correct, otherwise it is incorrect. We report overall accuracy in Table 2. In both mathematics and physics domains, Ax-Prover consistently surpasses both specialized prover models (DS-Prover and Kimina) and the standalone LLM baseline (Sonnet), highlighting its superior performance across the board.

On the Numina dataset, Ax-Prover largely outperformed all the baselines. Across all three NuminaMath-LEAN questions we benchmarked on, Ax-Prover scored 51% accuracy while DS-Prover (28%) and Kimina (31%) achieve similar performace, while Sonnet only gets 5% accuracy. Particularly notable is the performance of Ax-Prover on `Solved-H`, where it solves almost half of the problems, and on `Unsolved` (26%). Furthermore, due to autoformalization (see Section 4.3), some theorems are ill-posed: during testing, Ax-Prover encountered them, spotted the error, and reported it (see Appendix C).

On AA the gap in performance is striking, with Ax-Prover (64%) outperforming DS-Prover by 40%. Both Kimina (13%) and Sonnet get a very poor performance (8%). This is expected, as the AA dataset is largely out-of-distribution for DS-Prover, Sonnet, and Kimina, which are trained primarily on Mathlib — covering only a minimal subset of abstract algebra — or on undergraduate competition-level math problems.

On the QT dataset, Ax-Prover achieves an almost perfect performance (96%). DS-Prover also performed strongly (61%), while Kimina lagged significantly (57%), and Sonnet drops to 40%. To showcase the differences between the models, let's consider the proofs that quantum observables are Hermitian matrices (full proofs available in Appendix D.1). DeepSeek misused the Hermitian field, misunderstanding its type, while Sonnet made a more sophisticated effort but encountered a rewrite pattern mismatch, which highlights its difficulties in managing Lean environment. In contrast, Ax-Prover succeeded through a systematic approach, explicitly applying the Hermitian property to diagonal elements, using the definition of conjugate transpose, and connecting it to the fact that a complex number

---

[3]Each call may produce code or a tool invocation – for example, a call to `read_file` or `diagnostic_messages`.

[4]We could not test the baselines with pass@>1 due to budget restrictions.

equal to its conjugate is real. The case highlights that successful formal theorem proving requires careful, step-by-step reasoning, a solid grasp of type theory, and familiarity with library theorems – demonstrating that clarity and correctness outweigh clever shortcuts in formal verification.

These results highlight two key limitations of current approaches: general-purpose LLMs alone cannot effectively manage the Lean environment or leverage its tools, while specialized provers fail to generalize beyond their narrow training domains. The fact that Ax-Prover, built on Sonnet, nearly doubles the performance of the standalone LLM demonstrates that combining agentic reasoning with Lean tool integration is essential for robust theorem proving across domains.

### 5.3 ANALYSIS OF TACTIC USAGE

The large performance gap between Ax-Prover and standalone Sonnet underscores the importance of Lean tools. We measured tool usage over 100 runs on the challenging Numina `unsolved` subset, and found that the Prover makes an average of 100.76 tool calls per run.

However, one might still ask: do frequent tool calls actually enhance proofs quality and variety? To answer this question, we check the tactics used in the proofs by Ax-Prover, Kimina, and DeepSeek. (full stats in Table 3). While the three models share 28 tactics, Ax-Prover uses 10 tactics not employed by DS-Prover or Kimina, whereas the specialized provers use only one tactic absent in Ax-Prover. This supports our hypothesis that integrating frontier LLMs with Lean tools enhances both creative exploration and proof construction. Beyond performance, tool usage mirrors human Lean development, where interactive debugging and iterative proof refinement are central.

### 5.4 DEPLOYMENT ANALYSIS

As stressed in Section 1, deployment complexity is critical when using AI models. Here we compare models under this aspect. DeepSeek-Prover-V2-671B and Kimina-Prover-72B require GPU-accelerated, high-spec machines and are not available through model as a service (MaaS) providers – for instance, while DeepSeek-Prover-V2-671B was previously hosted by Novita (deepseek ai, 2025), this endpoint now redirects to the general DeepSeek-V3 model. We hosted DeepSeek and Kimina on Google Cloud: DeepSeek on an A3 Ultra VM with eight H200-141GB GPUs, and Kimina on an A2 High GPU VM with eight A100-40GB GPUs.

Deployment is burdensome and demands MLOps expertise: users must match hardware specs, configure distributed runtimes, debug serving issues, and contend with scarce GPU availability. Cloud providers enforce strict quotas and long queues for H100/H200 GPUs, hindering reproducibility even for well-funded teams.

In contrast, Ax-Prover relies only on making Anthropic API calls within an agentic loop, requiring no infrastructure beyond basic client access. It can be executed locally on a client machine or remotely in a lightweight container, enabling hundreds of problems per dataset to be evaluated without any engineering burden.

On monetary costs, running DS-Prover and Kimina on 1000 datapoints cost approximately $300 and $2000, respectively, while Ax-Prover cost about $4000. At first glance, our approach appears more expensive, but only because we evaluate specialized models with pass@1. Had we followed the common practice of running them with pass@32 or even pass@8192 (!), the cost of these specialized models would have far exceeded ours. Moreover, general-purpose LLMs are on a rapid trajectory of improvement: each new generation delivers stronger reasoning at lower cost, suggesting that the relative efficiency of Ax-Prover will only increase over time.

The deployment and cost barriers of specialized models also help explain why they have not achieved widespread use beyond benchmark settings such as IMO-style mathematics problems. For most researchers, the need to manage specialized hardware, navigate GPU quotas, and bear high costs makes these systems effectively unusable in practice. These barriers are eliminated with Ax-Prover. In the following section, we show how Ax-Prover can be directly used by domain experts, highlighting its potential as a researcher-friendly tool for real mathematical and scientific verification tasks.

## 6 USE CASE: RESEARCHER-FRIENDLY VERIFICATION

Besides being easily accessible, Ax-Prover supports interactive, collaborative workflows. Unlike large-scale provers that attempt to complete a proof in one shot, Ax-Prover allows researchers to inspect intermediate states, guide proof direction, and incorporate domain knowledge. This makes it not just a backend system, but a partner in mathematical and scientific reasoning. We provide a concrete example below.

We applied Ax-Prover to a recent cryptography result, *A New Algorithm for Computing Branch Number of Non-Singular Matrices over Finite Fields* (Mishra et al., 2024) (see the full case study in Appendix F). The task involved formalizing the paper's statments in Lean, and verifying the main claim. One of our mathematics domain experts collaborated with Ax-Prover to structure the proof, validate lemmas, and complete the verification all locally on their own laptop. Ax-Prover not only supported the process by checking intermediate lemmas and guiding proof strategies, but also revealed an error in the original approach – showing that it can not only reproduce known arguments, but also advance the state of knowledge. The whole process lasted two working days and resulted in over 2,000 lines of lean code and the resulting lean blueprint can be made available upon request.

Our case study marks one of the first concrete examples of AI systems directly assisting professional mathematicians. However, without Ax-Prover such successes currently require massive computational resources. For comparison, consider the formalization of the Prime Number Theorem (PNT): Terence Tao and Alex Kontorovich initiated the Lean translation, but the project was only completed *weeks later* by Math, Inc.'s Gauss agent running on Morph.AI's Infinibranch cluster (Math Inc., 2025). While ultimately successful, this was a massive engineering effort for a well-known proof of comparable difficulty to our cryptography case study.

This comparison underscores the contribution of Ax-Prover in enabling fast and efficient verification of research-level proofs of new results. Crucially, it acts as a teammate, by exposing intermediate reasoning, accepting guidance, and enforcing rigor.

# 7 CONCLUSIONS

We introduce **Ax-Prover**, a novel agentic workflow that combines the broad reasoning capabilities of general-purpose LLMs with the formal rigor of Lean's proof environment. Our system addresses three major limitations of current specialized provers: (i) limited generalizability and rapid obsolescence as libraries like `mathlib` evolve, (ii) high engineering and maintenance costs, and (iii) inability to leverage external tools or collaborate effectively with human experts.

Evaluations across multiple domains show that Ax-Prover consistently outperforms both specialized provers (DeepSeek-Prover, Kimina-Prover) and a standalone LLM baseline (Sonnet). On NuminaMath-LEAN it achieved higher overall success rates, including problems in the `Unsolved` split, and the gains were even more pronounced on our two new datasets, **AbstractAlgebra** and **Quantum Theorems**. These results highlight Ax-Prover's superior generalization, adapting to novel domains beyond its training data where specialized systems' performance worsens.

We attribute this success to our multi-agent architecture and its tight integration with Lean tools via the `lean-lsp-mcp` protocol. By iteratively editing proofs, inspecting goals, and diagnosing errors, Ax-Prover behaves like a cautious mathematician, systematically exploring and verifying each step. The frequency and effectiveness of tool use in our experiments confirm their essential role in improving proof quality and enabling human-like debugging.

Furthermore, in our case study on cryptography, we showcase the collaboration capabilities of Ax-Prover. One of our mathematics domain experts worked side-by-side with it, using it not just as an automated prover but as a partner for structuring arguments, validating intermediate lemmas, and diagnosing proof failures. This interaction demonstrates how Ax-Prover can adapt to expert guidance, accelerate verification workflows, and even surface errors in the original reasoning.

In addition, we contribute two new formalized datasets: **AbstractAlgebra**, which focuses on research-level mathematics, and **Quantum Theorems**. These benchmarks provide new testbeds for cross-domain reasoning in future agents.

Looking ahead, we plan to (i) enhance agentic performance with memory and parallelized agents to handle more complex, multi-step proofs; (ii) broaden the scope of our formalization efforts to applied scientific and engineering domains; and (iii) develop a more collaborative framework for human-agent interaction, enabling experts to guide agents on the most challenging problems.

## 7.1 ETHICS STATEMENT

All authors have read and adhere to the ICLR Code of Ethics.[5] Potential ethical considerations relate to the use of large language models for formal mathematics. First, compute costs and carbon footprint are a concern: although our approach requires API calls to frontier models, it avoids the extreme GPU usage required by specialized provers.

---

[5]https://iclr.cc/public/CodeOfEthics

Second, reproducibility and transparency are prioritized: all datasets used (NuminaMath-LEAN, AbstractAlgebra, and QuantumTheorems) are either publicly available or provided in our supplementary materials, and details of our setup are described in the Appendix. There are no other specific ethical concerns that we feel must be highlighted at this stage; however, we acknowledge the importance of ongoing evaluation of the societal and ethical implications as this technology is applied.

### 7.2 REPRODUCIBILITY STATEMENT

We have taken multiple steps to ensure reproducibility. Our datasets are either publicly available or attached in our supplementary materials (NuminaMath-LEAN, AbstractAlgebra, and QuantumTheorems), with details of their construction in Appendix B. All experimental settings, including pass@$k$ values, timeouts, and model configurations, are specified in Section 5 and Appendix E. We also describe our agentic workflow (Section 3), provide our code in the supplementary materials, provide example Lean proofs and failure cases (Appendix F), and document missing `mathlib` lemmas. Together, these materials are intended to allow researchers to fully reproduce our results and extend our experiments.

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

# A TOOLS

## A.1 FILE SYSTEM

Full list of File system tools:

- `read_file`
- `read_multiple_files`
- `write_file`
- `edit_file`
- `create_directory`
- `list_directory`
- `list_directory_with_sizes`
- `directory_tree`
- `move_file`
- `search_files`
- `get_file_info`
- `list_allowed_directories`

# B DATASETS

## B.1 ABSTRACT ALGEBRA

### B.1.1 DATASET GENERATION

We used a basic pipeline to build the abstract algebra dataset. First, we extracted all raw text from PDFs of material from *Abstract Algebra* textbooks by Dummit and Foote (Dummit & Foote, 2004) and Judsen (Judson, 2020) using Mistral's API. We then processed the raw text by using Claude-sonnet-3.7 to extract a list of natural language math statements. These natural language math statements contained exercises, derivations, lemmas, propositions, and theorems from the text.

Next, we used a Claude-sonnet-3.7 agent to autoformalize each of these natural language statements. To ground the formalization in Mathlib and prevent the agent from reinventing definitions, we passed the agent a lean file at the start of the process containing relevant definitions for that section, e.g. *dihedral groups*, *roots of unity*, or the *field extension* $\mathbb{Q}(\sqrt{2})$. The agent could reference these definitions and was required to add each formalized statement directly to this file, but explicitly prohibited from introducing new definitions. The agent generated the top 3 lean formal statements for each natural language statement and refined each attempt up to 3 times with feedback from the lean compiler. We then built the dataset by retaining only those pairs of natural language and formal language statements that corresponded to exercises from the source texts.

### B.1.2 EXAMPLE

This is an example proof of the triangle inequality for norms, which is fundamental in the mathematical structure of quantum state spaces.

```
import Mathlib

-- Variables for dihedral group
variable {n : } {i : }
local notation D => DihedralGroup n
local notation r => DihedralGroup.r (1 : ZMod n)
local notation s => DihedralGroup.sr (0 : ZMod n)

/--Use the generators and relations to show that every element of D not a power of r has order 2. -/
theorem exercise_3_part1 {x : D} (h : x = s * r^i) : orderOf x = 2 := by
  sorry
```

### B.2 QUANTUM-THEOREMS

#### B.2.1 DATASET GENERATION

The dataset was generated through an iterative human-in-the-loop process combining automated proof synthesis with expert curation. An automated coding agent first generated formal statements and proof attemps for all 134 quantum theorems, producing both complete proofs and partial derivations. A quantum physics expert then reviewed each statement, proof, identifying gaps, correcting errors, and standardizing operator definitions to ensure that each question was well formed and solvable. The final dataset replaces these proofs with `sorry` statements.

#### B.2.2 EXAMPLE

This is an example proof of the triangle inequality for norms, which is fundamental in the mathematical structure of quantum state spaces.

```
import Mathlib.Analysis.InnerProductSpace.Basic
import Mathlib.LinearAlgebra.UnitaryGroup
import Mathlib.LinearAlgebra.Matrix.Hermitian
import Mathlib.LinearAlgebra.Matrix.Trace
import Mathlib.Data.Complex.Basic
import Mathlib.Analysis.InnerProductSpace.PiL2

/-!
# Observable Linear Combination with Real Coefficients
-/

/-- Quantum state: normalized vector in Hilbert space (from KG) -/
def QuantumState (n : ) : Type :=
  { : EuclideanSpace  (Fin n) //  = 1}

/-- Observable: Hermitian matrix (from KG) -/
def Observable (n : ) : Type :=
  {A : Matrix (Fin n) (Fin n)  // A.conjTranspose = A}

theorem observable_real_linear_combination {n : } [NeZero n]
    (A B : Observable n) (  : ) :
     (C : Observable n), C.val = ( : )  A.val + ( : )  B.val := by
    sorry
```

## C   DETECTED AUTOFORMALIZATION ERROR

As noted in Section 5.2, 19.7% of Numina's problems were generated using autoformalization models. While these pipelines enable large-scale dataset construction, they occasionally produce ill-posed theorems that cannot be satisfied in Lean.

During evaluation, Ax-Prover successfully identified – and proved the contrapositive – of such a case.

```
import Mathlib

theorem number_theory_3098 (p1 p2 p3 p4 : ) (hp1 : p1.Prime) (hp2 : p2.Prime)
    (hp3 : p3.Prime) (hp4 : p4.Prime) (h1 : p1 < 100) (h2 : p2 < 100) (h3 : p3 < 100)
    (h4 : p4 < 100) (h5 : p1  p2) (h6 : p1  p3) (h7 : p1  p4) (h8 : p2  p3)
    (h9 : p2  p4) (h10 : p3  p4) (h11 : p1 = 1  p1 = 2  p1 = 3  p1 = 4  p1 = 5  p1 = 6  p1 = 7  p1 =
    (h12 : p2 = 1  p2 = 2  p2 = 3  p2 = 4  p2 = 5  p2 = 6  p2 = 7  p2 = 9)
    (h13 : p3 = 1  p3 = 2  p3 = 3  p3 = 4  p3 = 5  p3 = 6  p3 = 7  p3 = 9)
    (h14 : p4 = 1  p4 = 2  p4 = 3  p4 = 4  p4 = 5  p4 = 6  p4 = 7  p4 = 9)
    (h15 : p1  p2  p1  p3  p1  p4  p2  p3  p2  p4  p3  p4) :
    p1 + p2 + p3 + p4 = 190 := by  sorry
```

The first line of the proof sketch for this problem was

*This theorem has contradictory premises: the sum must be 17, not 190.*

Upon inspection, it is clear that 4 natural numbers belonging to the set $\{2, 3, 5, 7\}$ cannot sum to 190. As an additional excercise, we changed

```
p1 + p2 + p3 + p4 = 190 := by  sorry
```

to

```
p1 + p2 + p3 + p4 != 190 := by  sorry
```

changing the original theorem statement to prove the negation which Ax-Prover was able to do, thus proving that the original theorem was not provable. This behavior highlights two strengths of the agentic loop:

1. **Robustness to noise.** The agent does not blindly attempt to complete invalid statements but can detect contradictions early.

2. **Transparency.** By surfacing diagnostic messages directly from Lean, Ax-Prover provides clear evidence that the statement is ill-posed, enabling researchers to filter or repair dataset items.

We see this as an important complement to raw accuracy metrics: beyond solving valid theorems, a reliable prover should be able to recognize when formalization errors make a proof impossible.

## D PROOFS

### D.1 QUANTUM-THEOREMS ANALYSIS EXAMPLE

We present here the attempts to prove that quantum observables are Hermitian matrices in Lean 4, specifically proving the theorem:

$$\text{For an observable } A : \text{Observable } n, \quad A.\text{matrix}^{\dagger} = A.\text{matrix} \qquad (1)$$

where $A^{\dagger}$ denotes the conjugate transpose (Hermitian conjugate) of matrix $A$.

### D.1.1 DEEPSEEK'S ATTEMPT (FAILED)

DeepSeek's approach was fundamentally flawed from the start:

```
theorem observable_diagonal_real {n : } (A : Observable n) (i : Fin n) :
    (A.matrix i i).im = 0 := by
  have h := A.hermitian
  have h := congr_arg (fun x => x i i) h
  simp [Matrix.ext_iff, Matrix.conjTranspose_apply, Complex.ext_iff] at h
  <;>
    aesop
```

**Critical Error:**

```
tactic 'aesop' failed, made no progress
```

**Analysis:** DeepSeek attempted to use `A.hermitian` directly as a proof, but this field has type `matrix.IsHermitian`, not the required equality type. The error message indicates that the tactic `aesop` failed because there was a fundamental type mismatch. This shows a misunderstanding of Lean's type system. The `hermitian` field is a proof that the matrix satisfies the Hermitian property, but it's not directly the equality we need to prove.

### D.1.2 SONNET'S ATTEMPT (FAILED)

Sonnet made a more sophisticated attempt but still failed:

```
theorem observable_diagonal_real {n : } (A : Observable n) (i : Fin n) :
    (A.matrix i i).im = 0 := by
  have h := A.hermitian
  rw [Matrix.IsHermitian.ext_iff] at h
  specialize h i i
  rw [Matrix.conjTranspose_apply, Matrix.star_apply] at h
  exact Complex.ext_iff.mp h.symm |>.2
```

**Critical Error:**

```
Tactic `rewrite` failed: Did not find an occurrence of the pattern
  Matrix.conjTranspose ?M ?j ?i
in the target expression
  star (A.matrix i i) = A.matrix i i
```

**Analysis:** Sonnet understood that it needed to work with the Hermitian property and tried to extract information about diagonal elements, but made an error in pattern matching during the rewrite step. The approach was on the right track but executed incorrectly.

### D.1.3 AX-PROVER'S ATTEMPT (SUCCESSFUL)

Ax-Prover succeeded with a systematic, step-by-step approach:

```
theorem observable_diagonal_real {n : } (A : Observable n) (i : Fin n) :
    (A.matrix i i).im = 0 := by
  -- Step 1: Use the fact that A is Hermitian
  have h1 : A.matrix.conjTranspose = A.matrix := by
    exact A.hermitian

  -- Step 2: Apply this to diagonal element (i,i)
  have h2 : (A.matrix.conjTranspose i i) = (A.matrix i i) := by
    rw [h1]

  -- Step 3: By definition of conjugate transpose
  have h3 : (A.matrix.conjTranspose i i) = star (A.matrix i i) := by
    exact Matrix.conjTranspose_apply _ _ _

  -- Step 4: Combine to get star (A.matrix i i) = A.matrix i i
  have h4 : star (A.matrix i i) = A.matrix i i := by
    rw [ h3, h2]

  -- Step 5: A complex number equals its conjugate iff it's real
  have h5 : (A.matrix i i).im = 0 := by
    have : (starRingEnd ) (A.matrix i i) = A.matrix i i := h4
    exact Complex.conj_eq_iff_im.mp this

  exact h5
```

## E TACTICS

In Table 3 we report the tactics used by each model.

| Tactic | Ax-agent | DeepSeek | Kimina |
|---|---|---|---|
| apply | X | X | X |
| assumption | X | X | X |
| by_cases | X | X | X |
| calc | X | X | X |
| cases | X | X | X |
| change | X | | |
| classical | X | X | |
| congr | X | X | X |
| constructor | X | X | X |
| contradiction | X | X | X |
| decide | X | X | |
| exact | X | X | X |
| exact_mod_cast | X | X | X |
| exfalso | X | X | X |
| ext | X | X | X |
| funext | | X | X |
| generalize | X | | |
| induction | X | X | X |
| injection | X | | |
| intro | X | X | X |
| intros | X | | |
| left | X | | X |
| native_decide | X | | X |
| norm_cast | X | ? | |
| obtain | X | X | X |
| omega | X | X | X |
| push_cast | X | | |
| rcases | X | X | X |
| refine | X | X | X |
| replace | X | | X |
| rfl | X | X | X |
| right | X | | X |
| rintro | X | X | X |
| rw | X | X | X |
| rwa | X | | X |
| show | X | | X |
| simp | X | X | X |
| simp_all | X | X | X |
| simpa | X | X | X |
| specialize | | | X |
| subst | X | X | X |
| subst_vars | | X | |
| suffices | X | | |
| trans | X | | |
| unfold | X | | |

Table 3: Tactics used by Ax-agent, DeepSeek, and Kimina. An "X" indicates the model uses the tactic.

## F    CASE STUDY: VERIFYING MATH IN CRYPTOGRAPHIC PAPERS

In this case study, we illustrate how one of our researchers used Ax-Prover to verify correctness of mathematical results used in cryptographic research.

As a concrete example, we focus on the recent (May 2024) cryptographic paper *A New Algorithm for Computing Branch Number of Non-Singular Matrices over Finite Fields* from arXiv (Mishra et al., 2024). This work introduces a novel algorithm for computing the *branch number* – a fundamental metric used to assess the strength of block ciphers

such as AES (National Institute of Standards and Technology, 2001), PRINCE Borghoff et al. (2012), and Grøst Gauravaram et al. (2008).

The paper begins with **Theorem 1**, which offers an alternative characterization of the branch number. Traditionally, for an square invertible $n$ x $n$ matrix $M$ of order $n > 1$ over a finite field $\mathbb{F}_q$ of order $q$, the branch number is defined as

$$\mathcal{B}(M) = \min \left\{ w_h(x) + w_h(Mx) : x \in \mathbb{F}_q^n \text{ where } x \neq 0 \right\}$$

where $w_h(x)$ is the Hamming weight (the number of nonzero entries in $x$). Theorem 1 gives an alternate definition of the branch number that is more amenable to computation than the classical version.

**Theorem 1.** *The branch number of an invertible matrix $M \in M_n(\mathbb{F}_q)$ is given as*

$$\mathcal{B}(M) = \min \left\{ \min \left\{ h(M, x), h(M^{-1}, x) \right\} \mid x \in \mathbb{F}_q^n, 1 \leq w_h(x) \leq \left\lfloor \frac{n+1}{2} \right\rfloor \right\},$$

*where* $h(M, x) = w_h(x) + w_h(Mx)$.

For cryptographers, this makes a practical difference: it enables fast evaluation of candidate matrices when designing new lightweight or high-performance ciphers. The authors demonstrate in (Mishra et al., 2024, Theorem 4) that their algorithm achieves significant complexity gains over the naive $O(n^2 q^n)$ approach for finite fields of order $q \geq 4$ and square matrices of order $n \geq 4$.

### F.1 FORMALIZE: SINGLE STEP

To formally verify the math in this paper, we used an autoformalization agent to formalize statements, and then upon verifying that the formalization was correct we passed those statements to Ax-Prover.

To illustrate this process, we show the process of proving one step in the paper – the full lean certificate can provide upon request. The figure below shows the current verification state highlighted in green, while the next step awaiting verification appears in yellow.

Note that for the second term of the right-hand side of Equation (2), $h(M, x) = w_h(x) + w_h(Mx) > 2 \left\lfloor \frac{n+1}{2} \right\rfloor + 1 \geq n+1$. However, we know that the upper bound for $\mathcal{B}(M)$ is $n + 1$. Thus, we conclude that the second term of the right-hand side of (2) will not contribute to the computation of the branch number.

Therefore, from (1) and (2), we have

$$\mathcal{B}(M) = \min \left\{ \min \left\{ h(M, x) \mid x \in \mathbb{F}_q^n, 1 \leq w_h(x) \leq \left\lfloor \frac{n+1}{2} \right\rfloor \right\}, \right.$$
$$\left. \min \left\{ h(M, x) \mid x \in \mathbb{F}_q^n, \left\lfloor \frac{n+1}{2} \right\rfloor < w_h(x) \leq n, w_h(Mx) \leq \left\lfloor \frac{n+1}{2} \right\rfloor \right\} \right\}.$$
$$(3)$$

Again, we note that

$$\left\{ h(M, x) \mid x \in \mathbb{F}_q^n, 1 \leq w_h(x) \leq \left\lfloor \frac{n+1}{2} \right\rfloor, w_h(Mx) \leq \left\lfloor \frac{n+1}{2} \right\rfloor \right\} \subseteq$$
$$\left\{ h(M, x) \mid x \in \mathbb{F}_q^n, 1 \leq w_h(x) \leq \left\lfloor \frac{n+1}{2} \right\rfloor \right\}.$$

## F.2 VERIFICATION: SINGLE STEP

We highlighted the relevant theorems or lemmas and prompted Ax-Prover to generate a complete proof for each selected statement. Ax-Prover iteratively refined the proof, checking each step for correctness until the system confirmed its validity. Unlike standard LLM output, this process ensured full mathematical soundness, with Ax-Prover automatically searching for relevant results, reading and updating the file, diagnosing errors, and managing proof goals.

Some expert human oversight was required on 2 steps to properly formalize statements passed to the prover, but beyond these isolated interventions, the system operated autonomously. After several iterations, it produced a fully verified proof of each step in just a few minutes. The following prompt illustrates this process for the example above:

> **Prompt**
>
> Prove this statement (with the formalized step above highlighted as context).

> **Response**
>
> The statement is a simple set-theoretic observation. We start with all low-weight vectors x and look at the values $w_H(x) + w_H(Mx)$, where $w_H$ is the Hamming weight and M is some matrix.
>
> - Without constraints: We consider all low-weight vectors.
> - With constraints: We only consider those low-weight vectors where the output M x also has low weight.
>
> Clearly, if you add an extra condition ("M x has low weight"), you can only make the set smaller or equal—never larger—because you're restricting the set of vectors you're looking at.
> The proof just formalizes this obvious fact:
>
> 1. Take any element in the constrained set.
> 2. By definition, it comes from some vector x that satisfies the extra condition.
> 3. But x also satisfies the conditions for the unconstrained set (because the unconstrained set has fewer requirements).
> 4. So the element belongs to the unconstrained set as well.
>
> This shows the constrained set is a subset of the unconstrained set.

We now present an example that required expert oversight. Specifically, the Ax-Prover was unable to verify the correctness of statement (3) shown below:

> Note that for the second term of the right-hand side of Equation (2), $h(M, x) = w_h(x) + w_h(Mx) > 2 \lfloor \frac{n+1}{2} \rfloor + 1 \geq n + 1$. However, we know that the upper bound for $\mathcal{B}(M)$ is $n + 1$. Thus, we conclude that the second term of the right-hand side of (2) will not contribute to the computation of the branch number.

Therefore, from (1) and (2), we have

$$\mathcal{B}(M) = \min \left\{ \min \left\{ h(M, x) \mid x \in \mathbb{F}_q^n, 1 \leq w_h(x) \leq \left\lfloor \frac{n+1}{2} \right\rfloor \right\}, \right.$$
$$\left. \min \left\{ h(M, x) \mid x \in \mathbb{F}_q^n, \left\lfloor \frac{n+1}{2} \right\rfloor < w_h(x) \leq n, w_h(Mx) \leq \left\lfloor \frac{n+1}{2} \right\rfloor \right\} \right\}. \tag{3}$$

To guide the Ax-Prover, an expert user provided the following prompt:

> **Prompt**
>
> You have defined the sets $B, C, D$ with $A = \min\big(B, \min(C, D)\big) \le n + 1$. Lets do this by cases: If $\min C > \min D$ and $\min D > n + 1$, then $\min(C, D) > n + 1$, so $\min\big(B, \min(C, D)\big) = \min(B)$. The other case is easier.

The Axiomatic Verifier MCP then successfully proved the statement using the provided hint:

> **Response**
>
> VERIFICATION COMPLETE - STEPS:
>
> 1. **Define the minima:** Let
> $$B := \min\{ w_H(x) + w_H(Mx) \mid x \in L\},$$
> $$C := \min\{ w_H(x) + w_H(Mx) \mid x \in H_1\},$$
> $$D := \min\{ w_H(x) + w_H(Mx) \mid x \in H_2\}.$$
> where $L =$ low-weight set, $H_1 =$ high-weight with $w_H(Mx) \le (n + 1)/2$, $H_2 =$ high-weight with $w_H(Mx) > (n + 1)/2$. Then
> $$A := \mathrm{Branchnumber}(M) = \min(B, \min(C, D)).$$
> 2. **Upper bound:** For invertible $M$,
> $$A \le n + 1.$$
> 3. **High–high set bound:** If $x \in H_2$, then
> $$w_H(x) + w_H(Mx) > n + 1 \quad \Longrightarrow \quad D > n + 1.$$
> 4. **Case analysis:**
>    - $C \le D \quad \Rightarrow \quad \min(C, D) = C \quad \Rightarrow \quad A = \min(B, C)$.
>    - $C > D \quad \Rightarrow \quad \min(C, D) = D > n + 1$. Since $A = \min(B, D) \le n + 1$, we get $B \le n + 1$ and thus $A = B = \min(B, C)$.
> 5. **Conclusion:**
> $$A = \min(B, C).$$
> The term $D$ never affects the branch number.

## F.3 CONSOLIDATION: COMPLETE CERTIFICATE

Once all steps were verified, they were integrated into a single proof, producing a final file that formally certifies the alternate definition is equivalent to the original, exactly as proposed in the paper.

### F.4  Correctness, Verification, and Takeaways

The outcome of our work is a formally verified certificate ensuring the correctness of Theorem 1. During the course of constructing this certificate, we identified a critical error in the original proof. Specifically, the issue arises in **Step 2** of the proof:

$$
\min\left\{ h(M,x) \mid x \in \mathbb{F}_q^n, \left\lfloor \frac{n+1}{2} \right\rfloor < w_h(x) \le n \right\}
$$

$$
= \min\left\{ \min\left\{ h(M,x) \mid x \in \mathbb{F}_q^n, \left\lfloor \frac{n+1}{2} \right\rfloor < w_h(x) \le n, w_h(Mx) \le \left\lfloor \frac{n+1}{2} \right\rfloor \right\},\right.
$$

$$
\left. \min\left\{ h(M,x) \mid x \in \mathbb{F}_q^n, \left\lfloor \frac{n+1}{2} \right\rfloor < w_h(x) \le n, w_h(Mx) > \left\lfloor \frac{n+1}{2} \right\rfloor \right\} \right\}.
$$

$$
(2)
$$

Here, the authors fail to ensure that the sets over which they take minima are non-empty. For example, in the simplest case where $M = I$ (the identity matrix), the middle term reduces to

$$
\min\left\{ h(M,x) \mid x \in \mathbb{F}_q^n, \left\lfloor \frac{n+1}{2} \right\rfloor < w_h(x) \le n, w_h(x) \le \left\lfloor \frac{n+1}{2} \right\rfloor \right\}.
$$

In this case, the constraints
$$
\left\lfloor \tfrac{n+1}{2} \right\rfloor < w_h(x) \quad \text{and} \quad w_h(x) \le \left\lfloor \tfrac{n+1}{2} \right\rfloor
$$
are contradictory, so the underlying set is *empty*. Nevertheless, the original proof proceeds under the assumption that this minimum is well-defined, a subtle yet significant oversight.

This matters for two reasons:

1. **Logical correctness:** Reasoning about the empty set is problematic (all statements are *vacuously true*) which can lead to unsound conclusions. For example, let
$$
S = \{ x \in \mathbb{Z} \mid x = 3 \text{ and } x \text{ is even} \}.
$$
   Take $y \in S$, then $y = 3$ and $y$ is even, so this implies that 3 is even.

2. **Software implementation:** Computing the minimum of an empty set is undefined in standard programming environments and would trigger a runtime error if translated directly into code.

Our formal verification system flagged these issues because it could not establish the truth of the corresponding statements, revealing logical gaps in the proof. Nevertheless, the authors' final result remains correct despite the critical error in their proof.