# OpenReview forum: "Ax-Prover: Agentic LEAN Proving with LLMs and MCP-based Verifiers"
_ICLR.cc/2026/Conference — ICLR 2026 Conference Desk Rejected Submission_

### Official Review · Reviewer_RQ46 · 2025-10-27

**Soundness:** 2
**Presentation:** 2
**Contribution:** 2
**Rating:** 6
**Confidence:** 2

**Summary:**

This paper introduces Ax-prover, a multi-agent system for automated theorem proving. Ax-prover combines Large Language Models (LLMs) with MCP-based verification and validation tools. The experiments show that its performance outperforms current state-of-the-art (SOTA) provers across various domains.

**Strengths:**

This work combines LLM provers with tool-use capabilities, allowing the model to utilize functions like Lean search and Lean verification. This method is novel and holds significant value for the automated theorem proving (ATP) field.

Furthermore, their evaluation results show the system excels not only on standard math competition problems but also performs well in domains like abstract algebra and quantum theory. This breadth demonstrates the strong capabilities of their system.

**Weaknesses:**

The main weakness of this paper is its lack of technical detail. Only very high-level ideas are presented (e.g., the existence of three sub-agents).

Crucial information is missing, including:

1. How the models are trained.

2. The amount of data used for training.

3. Even the specific details of how their prover functions.

**Questions:**

Could you extend the experimental comparison to include more recent SOTA provers, such as Goedel-Prover-V2?

Could you release more specific details regarding your model training process and the overall system design?

---

### Official Review · Reviewer_gWSZ · 2025-11-01

**Soundness:** 2
**Presentation:** 3
**Contribution:** 2
**Rating:** 4
**Confidence:** 5

**Summary:**

This paper proposes Ax-Prover, a multi-agent system for automated theorem proving in Lean. The system employs an agentic approach that combines the broad reasoning capabilities of general-purpose Large Language Models (LLMs) with the formal rigor guaranteed by MCP toolsets integrated with the Lean environment.
The evaluation was conducted on the existing NuminaMath-LEAN dataset, as well as two new benchmarks newly constructed for this study: AbstractAlgebra (AA) and QuantumTheorems (QT). The experimental results show that Ax-Prover outperforms existing specialized provers (DS-Prover, Kimina) across all domains, demonstrating high generalization performance, especially on the new datasets.

**Strengths:**

1. **Construction of Novel Datasets**:
  A clear contribution of this research is the creation of two new Lean datasets (AA and QT) in new domains (abstract algebra and quantum theory). These complement existing benchmarks that are primarily focused on mathematics competitions. These datasets can serve as valuable resources for future automated theorem proving research to measure the generalization capabilities of models.
2. **Verification of Generalization Performance**:
  It is commendable that the paper uses these new datasets to demonstrate that while existing specialized provers lose performance outside their training domains, the proposed method shows domain adaptability.

**Weaknesses:**

1.  **Insufficient and Inappropriate Baseline Comparison**:
    * There are significant concerns about the validity of the paper's experimental evaluation. The Kimina Prover, used as a baseline, is arguably not a state-of-the-art (SOTA) model.
    * Regarding open-source models, comparisons against known stronger models, such as Goedel Prover v2, are missing.
    * The evaluation of closed models is also insufficient. Notably, the paper lacks a performance comparison against true SOTA models like Google's Gemini or OpenAI's GPT-5, which have reported extremely high performance in recent math competitions (e.g., IMO Grand Challenge). It is difficult to assert the effectiveness of the proposed method without demonstrating superiority over these models.

2.  **Constraints from Closed-Model Dependency**:
    * The reliance of the "Prover" agent on a specific closed API, Claude Sonnet 4, is a major limitation.
    * Importantly, this architecture fundamentally precludes research advancements that would be possible with open models, such as using reinforcement learning (RL) to further optimize the agent's proof strategies.

3.  **Lack of Novelty in Agent Architecture**:
    * The multi-agent configuration (Orchestrator, Prover, Verifier) itself does not present significant technical novelty compared to existing agent frameworks designed for proof assistant interaction.
    * The core idea of enabling a general-purpose LLM to solve tasks using external tools (in this case, Lean's MCP tools) is already well-established, making the paper's unique, fundamental contribution unclear.

**Questions:**

1.  **Regarding Baselines**: Why were more powerful, recent SOTA models (e.g., Goedel Prover v2, the latest Gemini, the latest GPT) not adopted as baselines? How would Ax-Prover's performance (and inference cost) be expected to compare against these models?
2.  **Regarding Closed-Model Dependency**: If a high-performance open-source model were used as the Prover agent instead of Claude Sonnet 4, how much of the performance would be maintained? Are there plans to reduce this dependency on closed models, or are there results from reproducibility experiments using open models?
3.  **Regarding Architectural Novelty**: Compared to existing LLM agent frameworks, what specific aspects of the proposed agent design are superior for the task of "theorem proving"? Please clarify the technical novelty more distinctly.

---

### Official Review · Reviewer_49ve · 2025-11-03

**Soundness:** 3
**Presentation:** 3
**Contribution:** 2
**Rating:** 4
**Confidence:** 4

**Summary:**

This paper presents Ax-Prover, a multi-agent system for automated theorem proving in Lean that bridges the gap between specialized theorem provers (narrow but Lean-integrated) and general-purpose LLMs (broad but lacking formal tools). The system consists of three agents: an Orchestrator (task management), a Prover (proof construction), and a Verifier (correctness checking). Ax-Prover leverages the lean-lsp-mcp protocol to give LLMs (specifically Claude Sonnet 4) direct access to Lean tools for inspecting goals, searching theorems, and verifying code.

The authors evaluate Ax-Prover on three datasets: NuminaMath-LEAN (competition mathematics), and two newly introduced benchmarks, AbstractAlgebra (graduate-level algebra) and QuantumTheorems (quantum physics). Results show Ax-Prover achieves 51%, 64%, and 96% accuracy respectively, outperforming both standalone Claude Sonnet 4 and specialized provers (DeepSeek-Prover-V2 and Kimina-Prover). The paper includes a case study verifying a cryptography result, demonstrating practical human-AI collaboration capabilities.

**Strengths:**

The authors successfully connect general-purpose LLMs to Lean via MCP, avoiding the need for specialized fine-tuning while maintaining formal rigor.

The multi-agent architecture is well-designed, with modular separation of concerns (orchestration, proving, verification) that enables independent optimization and extension.

Testing across competition mathematics, abstract algebra, and quantum physics demonstrates domain flexibility.

New benchmarks address a gap. AbstractAlgebra and QuantumTheorems target research-level mathematics and physics, expanding beyond competition-style problems.

The cryptography verification example (Appendix F) demonstrates practical utility and identifies an error in the original proof.

Table 3 and Section 5.3 show Ax-Prover uses 10 tactics not employed by specialized provers, suggesting enhanced exploration.

Deployment accessibility addresses a real point as specialized provers require expensive GPUs and engineering expertise while Ax-Prover only needs API access.

Tool usage analysis demonstrates average of 100.76 tool calls per run, providing evidence for the importance of Lean integration.

**Weaknesses:**

Major:

1. Critically flawed baseline comparison:
- Kimina results use pass@68 from training (acknowledged in footnote) vs. Ax-Prover's pass@1
- Table 2 notes "obtained during its RL training phase with, on average, pass@68"
- This makes the comparison meaningless, need fair pass@1 comparison
- Claims like "Ax-Prover largely outperformed all the baselines" are not supported


2. Severely limited experimental validation:
- Only 300/104,000 samples from NuminaMath-LEAN (0.3%)
- New datasets are small: 100 (AA) and 134 (QT) problems
- No evaluation on full miniF2F or Putnambench despite these being standard benchmarks
- Justification that "they contains similar problems types" is insufficient
- Only pass@1 evaluation (pass@k is standard in the field)


3. Poor generalization contradicts main claims:
- Appendix D (Figure 7) shows Ax-Prover and GRPO-hybrid both fail on miniF2F
- Paper claims Ax-Prover "adapts readily to novel areas" and existing systems "struggle to generalize"
- But miniF2F is the standard OOD benchmark, failure here is highly concerning
- The strong performance on new benchmarks might simply mean they're easier or closer to Claude's training distribution


4. Misleading cost analysis (Section 5.4):
- Compares 4000 dollars (Ax-Prover pass@1) vs. 300 dollars / 2000 dollars (DS / Kimina pass@1)
- Then argues specialized models "would have far exceeded ours" with pass@32+
- But this is unfair, comparing different evaluation protocols
- The paper uses pass@1 everywhere due to "budget restrictions" but then criticizes specialized models for needing pass@k
- Actual cost comparison should use same pass@k


5. Missing critical ablations:
- How much improvement comes from Claude Sonnet 4's base capability vs. the agentic workflow?
- What if you just give Claude Sonnet 4 access to diagnostic messages without full workflow?
- What percentage of proofs actually benefit from tool usage?
- How does performance scale with number of tool calls?


6. Dataset quality concerns:
- AbstractAlgebra: Semi-automatically generated, only 100 problems, no community validation
- QuantumTheorems: Only 134 problems, "human-in-the-loop process" but quality metrics unclear
- No difficulty analysis or comparison with existing benchmarks
- Risk that these datasets may be unintentionally biased toward Ax-Prover's strengths

Minor:

7. Case study limitations (Appendix F):
- Single example
- Required "expert human oversight on 2 steps"
- "whole process lasted two working days" with "over 2,000 lines of lean code"
- Not clear this is significantly better than manual formalization


8. Overclaiming deployment advantages:
- Valid point that specialized models need GPUs
- But Anthropic API calls also have rate limits, costs, and dependencies
- "hundreds of problems per dataset to be evaluated without any engineering burden" overstates ease
- Running 1000 problems at $4000 is not trivial for most researchers


9. Limited algorithmic novelty:
- Multi-agent systems are well-established
- The main contribution is engineering integration with MCP
- Workflow (identify -> sketch -> formalize -> solve -> verify) is standard human approach


10. Missing failure analysis:
- What are common failure modes?
- Which types of problems does Ax-Prover struggle with?
- When do tool calls help vs. hurt?
- What causes the Orchestrator to give up?

**Questions:**

Critical Questions:

1. Can you provide a fair pass@1 comparison with Kimina and DeepSeek-Prover on the same 300-sample NuminaMath subset? The current comparison with Kimina's pass@68 training results is not valid.

2. Why only 300 samples from NuminaMath-LEAN? With 104k problems available, this 0.3% sample seems insufficient. Can you evaluate on at least the full test set?

3. Can you explain the miniF2F failure? This directly contradicts claims about generalization. What specific challenges does miniF2F pose that your new benchmarks do not?

4. What is Ax-Prover's pass@k performance? Even pass@8 or pass@16 would be informative for understanding the method's reliability.

5. Can you run on full miniF2F and Putnambench? These are standard benchmarks and necessary for proper comparison with SOTA. The claim they are "too expensive" is undermined by spending $4000 on smaller datasets.

Important Questions:

6. What is the ablation of Claude Sonnet 4 base capability vs. agentic workflow? How does Sonnet 4 with just diagnostic messages compare to the full Ax-Prover workflow?

7. What percentage of successful proofs actually use Lean tools meaningfully? The average 100.76 tool calls includes many that might not contribute to success.

8. How does performance correlate with proof length/complexity? Are there systematic patterns in what Ax-Prover can vs. cannot prove?

9. What does "expert human oversight on 2 steps" mean in the case study? How much human intervention is generally needed? This is critical for assessing practical utility.

10. Can you provide inter-annotator agreement or expert validation for the new datasets? How do we know AbstractAlgebra and QuantumTheorems are well-formed and of appropriate difficulty?

Minor Questions:

11. How sensitive is performance to the 25-minute timeout? What about varying the number of Orchestrator-Prover-Verifier loops?

12. What is the distribution of tool usage across the different tool categories (Table 1)?

13. Can you provide a detailed comparison of proof strategies between Ax-Prover and specialized models beyond just tactic usage?

14. How does Ax-Prover perform when the base model is switched (e.g., GPT-4 instead of Claude Sonnet 4)?

15. What is the failure rate due to timeout vs. incorrect proofs vs. inability to find proof sketch?

---

### Official Review · Reviewer_LY2h · 2025-11-05

**Soundness:** 2
**Presentation:** 2
**Contribution:** 2
**Rating:** 2
**Confidence:** 4

**Summary:**

This paper introduced Ax-prover that consists of the prover, verifier and MCP tool usage. Their prover search for unfinished proofs, generate informal (natural language style) proof sketches, perform stepwise formalization, solve and also verify them. Their verifier verifies the generated proofs from the prover based on formal reasoning of Lean. Although their prover looks doing complex things, they indeed decompose the task and stepwisely generate formal code (lean) with the help of informal reasoning. They also introduced two new datasets: AbstractAlgebra(AT) and QuantumTheorems(QT) that cover from abstract algebra to quantum mechanics. Instead of widely-used formal reasoning datasets (miniF2F and PatnumBench), they did their experiments within these datasets, sacrificing the fair comparisons with existing strong formal reasoning models.

The critical problem of this paper is that  this paper ignores the existing approaches that utilize the prover and verifier in formal and informal math reasoning(Please closely see Baba et al. 2025 and DeltaProver). Indeed, this paper mentions lines of studies in agentic ai research out of formal math reasoning such as Gottweis et al. 2025 and Yamada et al. 2025 in Line 101 regardless of existing more relevant studies in formal math reasoning. This paper also fails to mention DeltaProver. In experiments, they compare their model with quite limited existing studies. Therefore the author failed to adequately position their study in lines of the formal math reasoning studies.

**Strengths:**

- S1: Proposal of new benchmark sets of AbstractAlgebra(AT) and QuantumTheorems(QT) that cover from abstract algebra to quantum mechanics.

**Weaknesses:**

- W1: The use of both prover and verifier is not novel and at least not the first idea in this paper. Please see Baba et al. (2025) and Zhou et al. (2025).
- W2: Lack of the comparison within the major evaluation benchmark (miniF2F and PatnumBench). They compare their model with three existing models mostly in their own configuration. This makes a fair comparison of Ax-Prover with many existing strong prover models (such as SeedProver and GoedelProver) almost impossible.
- W3: Limited effectiveness of the proposed benchmark sets: since the compared baselines are quite limited, the effectiveness and difficulty of the proposed datasets are not well presented.

**Questions:**

- Q1: Can you present the result if you apply your model to miniF2F and PatnumBench?
- Q2: Can you present more baseline model results with AbstractAlgebra(AT) and QuantumTheorems(QT)?

---

### Note · Program_Chairs · 2026-01-17
**Submission Desk Rejected by Program Chairs**

The following references in this submission do not refer to real documents and/or have major errors in bibliographic information:

 Josef Urban, Geoff Sutcliffe, Stefan Petrov, and Josef Vyskočil. Machine learning preselected proof steps. In Proceedings of the International Joint Conference on Artificial Intelligence (IJCAI), pp. 2046-2051, 2011.
Guillaume Lample and François Charton. Deep reinforcement learning for theorem proving. In International Conference on Learning Representations (ICLR), 2022.